# Recursive RX with Extended Multi-Attribute Profiles for Hyperspectral Anomaly Detection

**Fang He [1,†]****, Shuai Yan [1,2,†], Yao Ding [1,\*], Zhensheng Sun [1], Jianwei Zhao [1], Haojie Hu [1] and Yujie Zhu [1]**

[1] Xi'an Research Institute of Hi-Tech, Xi'an 710025, China
[2] College of Marxism, National University of Defense Technology, Wuhan 430019, China
* Correspondence: dingyao.88@outlook.com
† These authors contributed equally to this work.

**Abstract:** Hyperspectral anomaly detection (HAD) plays an important role in military and civilian applications and has attracted a lot of research. The well-known Reed–Xiaoli (RX) algorithm is the benchmark of HAD methods. Based on the RX model, many variants have been developed. However, most of them ignore the spatial characteristics of hyperspectral images (HSIs). In this paper, we combine the extended multi-attribute profiles (EMAP) and RX algorithm to propose the Recursive RX with Extended Multi-Attribute Profiles (RRXEMAP) algorithm. Firstly, EMAP is utilized to extract the spatial structure information of HSI. Then, a simple method of background purification is proposed. That is, the background is purified by utilizing the RX detector to remove the pixels that are more likely to be anomalies, which helps improve the ability of background estimation. In addition, a parameter is utilized to control the purification level and can be selected by experiments. Finally, the RX detector is used again between the EMAP feature and the new background distribution to judge the anomaly. Experimental results on six real hyperspectral datasets and a synthetic dataset demonstrate the effectiveness of the proposed RRXEMAP method and the importance of using the EMAP feature and background purity means. Especially, on the abu-airport-2 dataset, the AUC value obtained by the present method is 0.9858, which is higher than the second one, CRD, by 0.0198.

**Keywords:** hyperspectral anomaly detection (HAD); extended multi-attribute profiles (EMAP); hyperspectral image (HSI); Reed–Xiaoli (RX); background purity

## 1. Introduction

A hyperspectral image (HSI) provides a rich source of spectral and spatial information about the materials in the scene [1], and it has been widely applied in many remote sensing areas [2–4], including classification [5–9], clustering [10–12], unmixing [13–15], image denoising [16,17], band selection [18,19], change detection [20], and target detection [21–25] or anomaly detection [26–33]. Among these applications, anomaly detection (AD) plays a significant role in military surveillance [34], agriculture [35], mineral exploration [36], environmental monitoring [34], maritime rescue [37], and so on [27,38–41].

Anomaly detection does not need any prior information related to anomalous targets or background. A pixel can be viewed as an anomaly if it is different from its neighbors. Usually, the anomalous pixels have a low probability of occurrence. Since the spectral characteristics of each class are different, the spectral signature of an anomalous pixel differs from that of the background pixels. Hypespectral anomaly detection (HAD) aims to distinguish the anomaly class from the background class [38]. Over the past three decades, a lot of HAD methods have been proposed. The well-known Reed–Xiaoli (RX) [42] algorithm is the benchmark for anomaly detection in HSIs. Later, some variants of RX methods and non-RX-based methods have also been proposed.

The RX algorithm assumes that the background pixels follow a multivariate normal distribution. The probability density function of the multivariate normal distribution is

adopted to measure the probability of the test pixel to be part of the background. After a series of mathematical derivations, it can be found that the Mahalanobis distance between the spectral vectors of an input test pixel and its surrounding neighbors can be obtained by the resulting generalized likelihood ratio test [38,42]. The effectiveness of the RX algorithm has been validated in multispectral and hyperspectral images, and it is widely used in hyperspectral imaging applications [43].

Based on the RX algorithm, many variants have been proposed, e.g., the global RX (GRX) approach using the whole image to model the background and the local RX approach modeling the background with a small neighborhood of the test pixel [44]. Later, a nonlinear version of the RX algorithm, Kernel RX (KRX) [45], was developed, which uses the kernel theory to map the HSI into a high-dimension feature space and exploits the higher-order information among the spectral bands to complete detection. Following the similar idea, a modified KRX algorithm was devised assuming that the background class is a spherical covariance matrix for the background class [46]. Furthermore, a new cluster kernel RX [47] was proposed, which firstly divides the background pixels into clusters and then applies a fast eigenvalue decomposition algorithm to generate the anomaly detection index. Some other types of RX algorithms have also been developed. The regularized-RX [48] uses all HSI pixels to regularize the covariance matrix estimated. Weighted RX (WRX) [49] assigns a special weight to each pixel to model the background. The median–mean line RX (MML-RX) [26] detector uses the median–mean line (MML) metric to degrade the negative effects on the mean and covariance matrix caused by anomalies. The blocked adaptive computationally efficient outlier nominator (BACON) method iteratively updates a stable and robust background subset to suppress the contamination of the sample mean and covariance [50].

At the same time, many non-RX-based HAD algorithms have also been developed, e.g., some representation-based algorithms, which include the collaborative representation-based algorithms [51–54], the sparse representation-based algorithms [55–60], and the low-rank representation-based algorithms [38,39,61–64]. Li and Du [51] proposed a collaborative representation-based anomaly detection (CRD) algorithm, which is based on the perspective that the background pixels can be approximately represented by its surrounding pixels in a local dual window, while the anomalous pixels cannot. Chen et al. [55] proposed a sparse representation-based anomaly detection algorithm, assuming that a pixel in HSI lies in a low-dimensional subspace and can be represented as a sparse linear combination of the training samples. Xu et al. [38] proposed the low-rank and sparse representation (LRASR) algorithm, which is built on the separation of the anomaly part and background part. The background information is contained in the lowest rank representation of the HSI pixels. Sun et al. [61] proposed the low-rank and sparse matrix decomposition (LRaSMD) algorithm, assuming that the background image is low-rank, while anomalies are gross errors distributed through the an image scene sparsely. Zhang et al. [62] further proposed the low-rank and sparse matrix decomposition-based Mahalanobis distance (LSMAD) algorithm, which takes full advantage of the LRaSMD technique to set the background apart from the anomalies. LSMAD explores the low-rank prior knowledge of the background pixels to compute the background statistics and detect the anomalies according to the Mahalanobis distance.

In addition, there are other proposed methods aiming at improving the detection performance by purifying background. For example, the random-selection-based anomaly detector (RSAD) [37] randomly selects representative background pixels to compute background statistics through designing a sample random selection process, which can reduce the contamination of the background statistics by anomaly pixels. The Gaussian background purification (GBP) [65] technique obtains a clean local background by removing the low-probability pixels from the local window according to background data samples probability distribution. The background-purification-based (BPB) framework focuses on accurate background information estimation by considering the role of background estimation and suppression in anomaly detection [66]. The spatial density background

purification (SDBP) [28] method uses a density peak clustering (DP) algorithm to calculate the local density of pixels within the double window. Then, the local densities can be sorted into descending order, and the *m* pixels with the highest local density are selected. In this way, the potential abnormal pixels in the background can be effectively removed, which is helpful to obtain a purer background. Finally, the collaborative representation detector (CRD) is employed to complete anomaly detection.

In this paper, we focus on the RX-based algorithm. Most RX algorithms are based on the spectral features of hyperspectral images, ignoring the spatial characteristics. As HSI images contain rich spatial information, the performance of hyperspectral image processing can be improved by effectively using these spatial information. Many studies have been devoted to extracting the spatial features in the image analysis. Pesaresi and Benediktsson [67] presented the morphological profiles (MPs) to effectively model the spatial information, which was successfully employed in the classification of high-resolution panchromatic IKONOS images. Based on MPs, Dalla Mura et al. [68] presented attribute profiles (APs). APs modeled the spatial information more precisely with respect to MPs. As the input image can be processed according to many attributes, it can be defined with great flexibility. Then, the extended attribute profile (EAP) and the extended multi-attribute profile (EMAP) [69] were developed. EAP relied on the application of the APs to hyperspectral data. EMAP was a straightforward further extension of EAP to a multi-attribute scenario.

In this paper, we utilize the EMAP feature to represent the spatial structure information of HSI and propose the Recursive RX with EMAP (RRXEMAP) method to more accurately identify abnormal targets. The main contributions are as follows:

1   The extended multi-attribute profile (EMAP) is used to extract spatial features of hyperspectral images, which can improve the detection performance by combining the spatial information.
2   According to the goal of anomaly detection, the RX detector is adopted to remove the pixels that are more likely to be anomalies, which can purify the background and help estimate the background distribution.
3   The RX model is used on the purified image to detecte the anomaly.

This paper is organized as follows. Section 2 briefly reviews the related work. Then, the details of the proposed RRXEMAP are given in Section 3. Experimental results are discussed in Section 4. Finally, conclusions are presented in Section 5.

## 2. Related Work

The Reed–Xiaoli (RX) [42] algorithm assumes that the background pixels obey a multivariate Gaussian distribution and has two versions: global RX (GRX) and local RX (LRX) [44]. In this section, we review these two algorithms.

### 2.1. Global RX

The GRX algorithm uses the sample correlation matrix for the full hyperspectral image to detect anomalies, which has been widely used in signal and image processing [43]. Let $\boldsymbol{X} \in \mathbb{R}^{d \times n}$ denote the HSI data, where $d$ and $n$ are the number of dimensions and pixels, respectively. The GRX algorithm needs to distinguish the following two competing hypotheses:

$$\left\{ \begin{array}{l} H_0 : \boldsymbol{x} \sim N(\mu_b, \boldsymbol{C}_b) \\ H_1 : \boldsymbol{x} \sim N(\mu_s, \boldsymbol{C}_b) \end{array} \right. , \tag{1}$$

where $\mu_b$ and $\mu_s$ are the mean vector of the background and anomalous target, respectively. $\boldsymbol{C}_b$ is the covariance matrix of the background.

According to these hypotheses, anomalous targets can be determined by comparing with an appropriate threshold $\tau$ as follows:

$$H_0$$

$$s_i \lessgtr \tau. (i = 1, 2 \cdots, n). \tag{2}$$

$$H_1$$

If $s_i$ is larger than $\tau$, the $i$-th pixel is claimed to be an anomaly; otherwise, it belongs to the background. Here, $s_i$ is the anomaly score and can be calculated by:

$$
\begin{aligned}
s_i &= (\boldsymbol{x}_i - \mu)^T \left[ \frac{n}{n+1} \boldsymbol{C} + \frac{1}{n+1} (\boldsymbol{x}_i - \mu)(\boldsymbol{x}_i - \mu)^T \right]^{-1} (\boldsymbol{x}_i - \mu) \\
&\xrightarrow[n \to \infty]{} (\boldsymbol{x}_i - \mu)^T \boldsymbol{C}^{-1} (\boldsymbol{x}_i - \mu)
\end{aligned}
\tag{3}
$$

where $\mu$ and $\boldsymbol{C}$ are the mean vector and covariance matrix of the background distribution, respectively, which are determined by:

$$
\begin{aligned}
\mu &= \frac{1}{n} \sum_{i=1}^{n} \boldsymbol{x}_i \\
\boldsymbol{C} &= \frac{1}{n} \sum_{i=1}^{n} (\boldsymbol{x}_i - \mu)(\boldsymbol{x}_i - \mu)^T
\end{aligned}
\tag{4}
$$

### 2.2. Local RX

The LRX algorithm uses the local dual-window image to model the background, which is an important anomaly detection approach. When the anomalies are relatively small or only distinct from the local surroundings but buried in the global background, the GRX algorithm may fail to work [44]. The LRX detector can be viewed as a local anomaly detector because each pixel in the image has its own correlation matrix, which can be calculated by using a small set of neighboring pixels around the pixel under test. For each pixel, the LRX filter is computed using a square window of size $w \times w$ pixels, centered at pixel $\boldsymbol{x}$. The form of the dual window is shown in Figure 1. Then, the local covariance matrix $\boldsymbol{C}_{local}$ and mean $\mu_{local}$ can be calculated for every pixel $\boldsymbol{x}$.

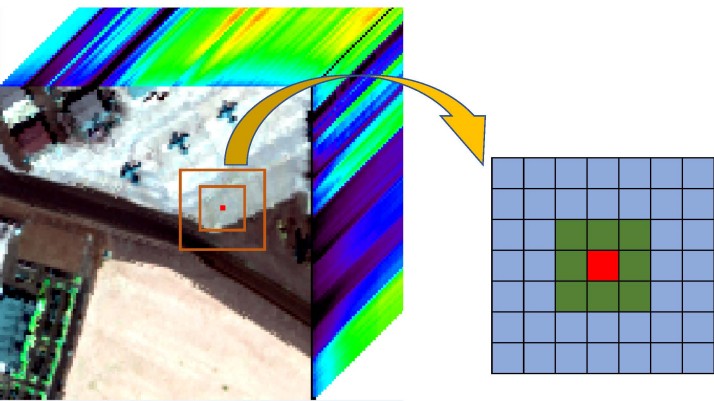

**Figure 1.** Description of the dual window.

The anomaly score of LRX can be specified as follows:

$$s_i^{LRX} = (i - \mu_{local})^T \boldsymbol{C}_{local}^{-1} (i - \mu_{local}). \tag{5}$$

As the covariance matrix $C_{local}$ is calculated by using the local window instead of the full image, LRX is a local approach estimation to complete anomaly detection. LRX processes each pixel of the image by using the local information of the data in the sliding window and applying the RX filter in the local area [43].

## 3. Proposed Method

In this section, we propose a new hyperspectral anomaly detection algorithm called recursive RX with extended multi-attribute profiles (EMAP) (RRXEMAP). Firstly, the EMAP method is utilized to extract the spatial features. Then, a new background purification method is proposed by adopting the RX detector. Finally, the RX method is used again to complete the final anomaly detection. The detail model is shown in Figure 2.

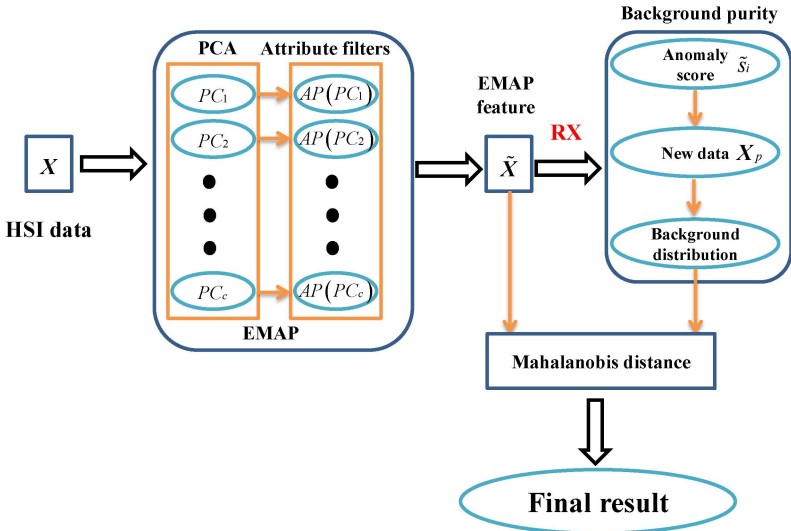

**Figure 2.** Description of RRXEMAP model.

### 3.1. Extended Multi-Attribute Profiles

The extended multi-attribute profile (EMAP) is based on the morphological attribute filters, which can extract spatial features through a multi-level analysis to better utilize the spatial information. Now, we have a brief review of the EMAP algorithm.

EMAP is based on the extended attribute profiles (EAPs), which can be computed on the $c$ PCs extracted from the original hyperspectral data. The definition of the EAP is:

$$EAP = \{AP(PC_1), AP(PC_2), \cdots, AP(PC_c)\}, \tag{6}$$

where APs (attribute profiles) are a multi-level decomposition of the input image based on the attribute filters [68]. The AP can be defined as a concatenation of a thickening AP, $\prod_{\phi_\lambda}$ and a thinning AP, $\prod_{\gamma_\lambda}$.

$$AP(f) = \prod_i : \begin{cases} \prod_i = \prod_{\phi_\lambda}, with\lambda = (n-1+i), \forall \lambda \in [1,n]; \\ \prod_i = \prod_{\gamma_\lambda}, with\lambda = (i-n-1), \forall \lambda \in [n+1, 2n+1]. \end{cases} \tag{7}$$

Since APs utilize different attributes to extract different information from the scene, the EAPs can be further evolved to the EMAP. The EMAP merges different EAPs in a single data structure, which can be defined as:

$$EMAP = \{EAP_{a1}, EAP'_{a2}, \cdots, EAP'_{am}\}, \tag{8}$$

where $a_i$ is a generic attribute and $EAP' = EAP \smallsetminus \{PC_1, \cdots, PC_c\}$.

In this paper, considering four different attributes: area of the regions, size of the regions, elongation of the regions, and homogeneity of the regions, an EAP is computed for each of these attributes. Then, the EMAP feature for each pixel can be constructed by concatenating all the attribute profiles. The EMAP feature matrix is represented as $\widetilde{X} \in \mathbb{R}^{d_4 \times n}$, where $d_4$ is related to the parameters of the attribute filters and the number of principal components. As in [69], each attribute can generate nine features. If we use three top principal component images, then $d_4 = 9 \times 4 \times 3 = 108$.

The procedures to generate EMAP feature can be summarized in Algorithm 1.

---

**Algorithm 1:** EMAP

---

**Input:** Hyperspectral data $X \in \mathbb{R}^{d \times n}$, the number of principal components $c$.

1. Generate the top $c$ principal component images, i.e., $PC_1, PC_2, \cdots, PC_c$ by the principal component analysis (PCA) technology.
2. Generate the extended attribute profiles (EAP) for each principal component image by the morphological attribute filters.
3. The EAP is further evolved to the EMAP with four different attributes: area of the regions, size of the regions, elongation of the regions and homogeneity of the regions.

**Output:** EMAP feature matrix $\widetilde{X} \in \mathbb{R}^{d_4 \times n}$, where $d_4 = 9 \times 4 \times c$.

---

### 3.2. Recursive RX Using Extended Multi-Attribute Profiles

Once the EMAP feature matrix $\widetilde{X} \in \mathbb{R}^{d_4 \times n}$ is obtained, the RX detector can be adopted to estimate the background distribution of the EMAP image. The background distribution is only determined by $\widetilde{\mu}$ and $\widetilde{C}$, which can be calculated in the following way:

$$
\begin{aligned}
\widetilde{\mu} &= \frac{1}{n} \sum_{i=1}^{n} \widetilde{X}_i \\
\widetilde{C} &= \frac{1}{n} \sum_{i=1}^{n} \left( \widetilde{X}_i - \widetilde{\mu} \right) \left( \widetilde{X}_i - \widetilde{\mu} \right)^T
\end{aligned}
\tag{9}
$$

Then, the anomaly score for the $i$-th test pixel of the EMAP image can be obtained as follows:

$$
\widetilde{s}_i = \left( \widetilde{X}_i - \widetilde{\mu} \right)^T \widetilde{C}^{-1} \left( \widetilde{X}_i - \widetilde{\mu} \right).
\tag{10}
$$

From Equation (**??**), we can find that the larger $\widetilde{s}_i$ is, the greater the probability that the corresponding data are an anomaly. Therefore, order $\widetilde{s}_i$ from smallest to largest and remain the top $n \times p$ pixels $X_p \in \mathbb{R}^{d_4 \times np}$, where $p$ is the percentage of the whole data. The new background distribution $\widetilde{\mu}_p$ and $\widetilde{C}_p$ can be calculated in the same way as Equation (9). In this way, a purer background can be obtained, which is helpful for anomaly detection.

After obtaining $\widetilde{\mu}_p$ and $\widetilde{C}_p$, the anomaly score can be acquired by calculating the Mahalanobis distance between the new background and the EMAP image.

$$
sp(j) = \left( j - \widetilde{\mu}_p \right)^T \widetilde{C}_p^{-1} \left( j - \widetilde{\mu}_p \right),
\tag{11}
$$

where $j \in X_p$.

Finally, setting an appropriate threshold $\tau_p$, the anomalous targets can be determined:

$$H_0$$

$$
sp_i \lessgtr \tau. (i = 1, 2 \cdots, n)
\tag{12}
$$

$$H_1$$

If $sp_i$ is larger than $\tau$, the $i$-th pixel is claimed to be an anomaly; otherwise, it belongs to the background cluster. The detailed algorithm can be summarized in Algorithm 2.

---

**Algorithm 2:** RRXEMAP algorithm

---

**Input:** Hyperspectral data $X \in \mathbb{R}^{d \times n}$, the number of principal components $c$.

1. Calculate the EMAP feature matrix $\widetilde{X} \in \mathbb{R}^{d_4 \times n}$ according to Algorithm 1.
2. Perform RX anomaly detection on $\widetilde{X}$ to obtain the anomaly score $\widetilde{s}_i$.
3. Sort $\widetilde{s}_i$ from smallest to largest.
4. Remain the top $n \times p$ pixels $X_p \in \mathbb{R}^{d_4 \times np}$, where $p$ is the percentage of the whole data.
5. Calculate the new background distribution $\widetilde{\mu}_p$ and $\widetilde{C}_p$.
6. Calculate the Mahalanobis distance between the new background and the EMAP image according to Equation (**??**) to obtain the anomaly score $sp(j)$.

**Output:** The anomaly score $sp$ for each pixel.

---

## 4. Experiments

In this section, we conduct several experiments on six real hyperspectral datasets and a synthetic dataset to evaluate the detection performance of the proposed RRXEMAP model. All experiments are implemented on a PC with i5-7200U @2.5GHz and 8GB RAM, MATLAB 2017a. The detailed descriptions of the seven datasets are as follows:

1. AVIRIS-I Dataset: This dataset was acquired by the Airborne Visible/Infrared Imaging Spectrometer (AVIRIS) in San Diego, CA, USA. The whole image scene has $400 \times 400$ pixels, with 224 spectral bands in wavelengths ranging from 370 to 2510 nm. The AVIRIS-I dataset is a region with a size of $120 \times 120$ pixels from the top left of this image, which was obtained from [38]. After removing the water absorption, low-signal-to-noise ratio, and poor-quality bands, only 189 bands remained in the experiments. In this scene, three airplanes composed of 58 pixels are regarded as anomalies, which need to be detected. The false color image and the corresponding ground truth map of the AVIRIS-I dataset are shown in Figures 3a and 3d, respectively.

2. AVIRIS-II Dataset: Compared with the AVIRIS-I dataset, the AVIRIS-II dataset is a region with a size of $100 \times 100$ pixels from the center of San Diego, which was obtained from [70]. In this image, three airplanes composed of 134 pixels are regarded as anomalies, which need to be detected. The false color image and the corresponding ground truth map of the AVIRIS-I dataset are presented in Figures 3b and 3e, respectively.

3. Cri Dataset: This dataset was obtained by the Nuance Cri hyperspectral sensor and acquired from [38], with a size of $400 \times 400$ pixels and 46 spectral bands in wavelengths ranging from 650 to 1100 nm. There are ten rocks represented by 2216 pixels regarded as anomalies to be detected in this image scene. The false color image and the corresponding ground truth map of the AVIRIS-I dataset are presented in Figures 3c and 3f, respectively.

4. abu-airport-2 Dataset: This dataset was acquired by the Airborne Visible/Infrared Imaging Spectrometer (AVIRIS) from ABU datasets [70], with a size of $100 \times 100$ pixels and 204 spectral bands. The two airports are regarded as anomalies in this image scene. The false color image and the corresponding ground truth map of the abu-airport-2 dataset are presented in Figures 4a and 4d, respectively.

5. abu-urban-1 Dataset: This dataset is an urban scene image from ABU datasets [70], with a size of $100 \times 100$ pixels and 204 spectral bands. Various vehicles and man-made targets are regarded as anomalies in this image scene. The false color image and the corresponding ground truth map of the abu-urban-1 dataset are presented in Figures 4b and 4e, respectively.

6. abu-urban-3 Dataset: This dataset is the same as the abu-urban-1 scene from ABU datasets [70]. The false color image and the corresponding ground truth map of the abu-urban-3 dataset are presented in Figures 4c and 4f, respectively.

7. Salinas-simulate Dataset: This dataset has a synthetic anomaly, which was obtained by using the target implant method on a different portion of the Salinas scene. The Salinas scene was collected over Salinas Valley, California [71]. The whole scene has $512 \times 217$ pixels, 16 classes, with 224 spectral bands. After removing the water absorption,

low-signal-to-noise ratio, and poor-quality bands, only 204 bands are valuable. The false color image consists of bands 70, 27 and 17 and the corresponding ground truth map is respectively provided in Figure 5a,b. The Salinas-simulate dataset covers a region with a size of $150 \times 200$ pixels and 204 bands from the Salinas scene, which was obtained from [71]. In this scene, a $150 \times 126$ binary mask image $M$ shown in Figure 5d has been constructed by generating six squares, which have sides measuring from 1 to 6 pixels arranged in a line. The six squares have been copied in reverse order and arranged in another line at close distance. The two lines have been rotated by an angle of approximatively $\pi/6$. The value of pixels inside the squares is 1, while that of the rest of the pixels in $M$ is 0. Next, a region $I$ is cropped from the Salinas scene with the same dimension as the mask. The modified image $I'$ can be built, which contains the implanted target as follows:

$$I'(i,j) = M(i,j) \cdot \phi(k) + (1 - M(i,j)) \cdot I(i,j) \tag{13}$$

where $\phi$ is a function, with a parameter $k \in [1,16]$, that returns a random pixel from the region of the Salinas scene having class $k$ according to the classification ground truth shown in Figure 5b. In this paper, $k = 14$, the false color image and the corresponding ground truth map of the Salinas-simulate dataset are presented in Figures 5c and 5d, respectively.

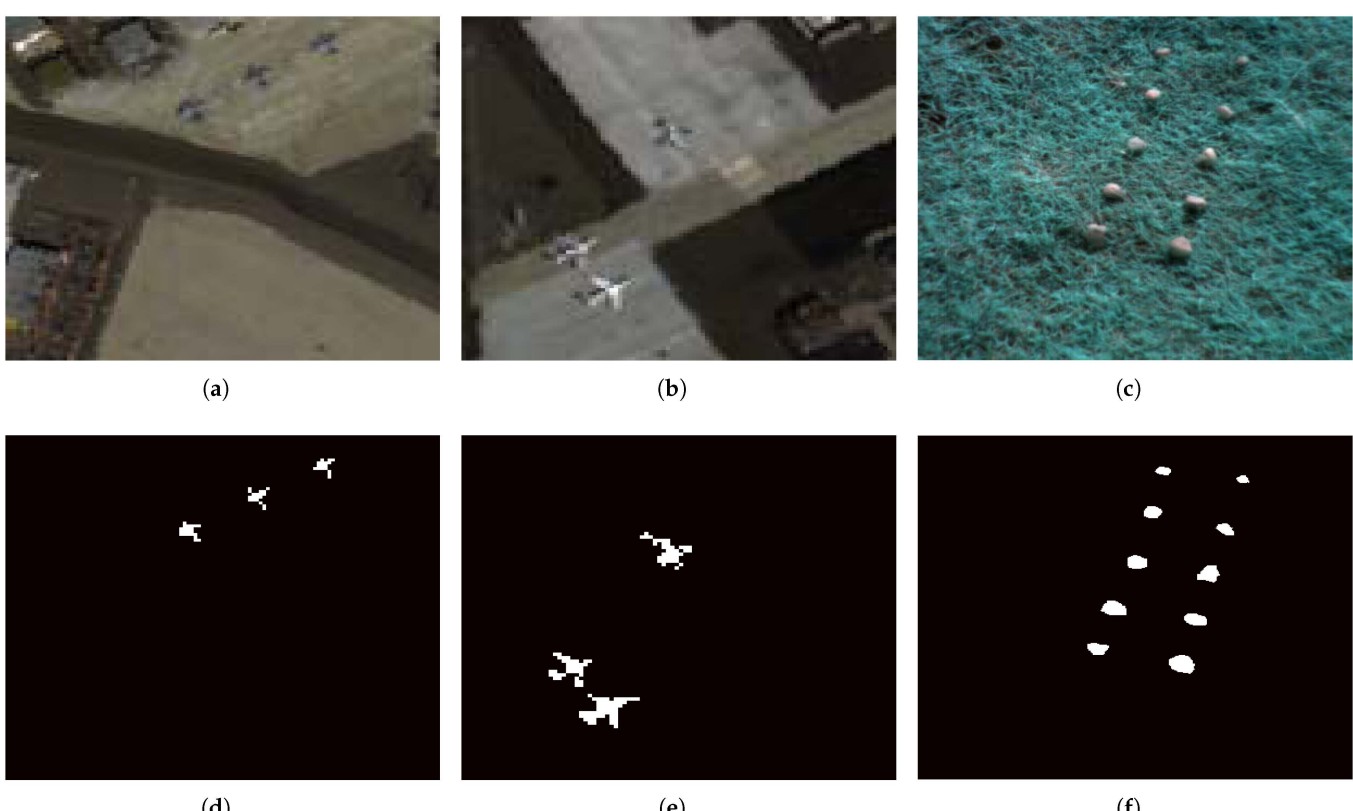

**Figure 3.** Image scene descriptions. (**a**) False color image of the AVIRIS-I dataset. (**b**) False color image of the AVIRIS-II dataset. (**c**) False color image of the Cri dataset. (**d**) The ground truth map of the AVIRIS-I dataset. (**e**) The ground truth map of the AVIRIS-II dataset. (**f**) The ground truth map of the Cri dataset.

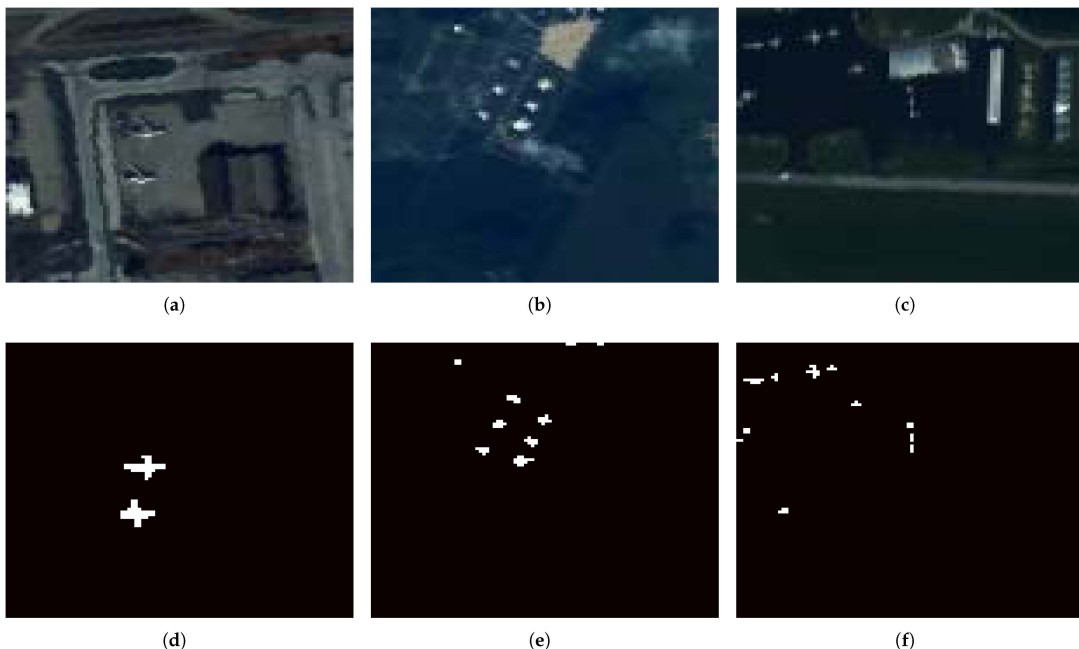

**Figure 4.** Image scene descriptions. (**a**) False color image of the abu-airport-2 dataset. (**b**) False color image of the abu-urban-1 dataset. (**c**) False color image of the abu-urban-3 dataset. (**d**) The ground truth map of the abu-airport-2 dataset (**e**) The ground truth map of the abu-urban-1 dataset. (**f**) The ground truth map of the abu-urban-3 dataset.

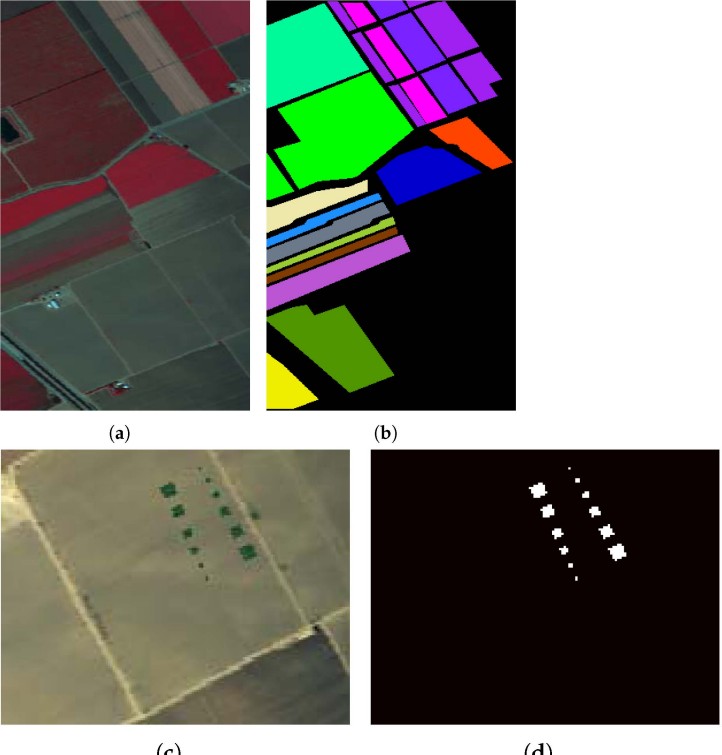

**Figure 5.** Image scene descriptions. (**a**) False color image of the Salinas dataset. (**b**) The ground truth map of the Salinas dataset. (**c**) False color image of the Salinas-simulate dataset. (**d**) The ground truth map of the Salinas-simulate dataset.

### 4.1. Evaluation Methods

In our experiments, to have an intuitive evaluation metric, we use the color detection map as the qualitative evaluation metric. The receive operating characteristic (ROC) curve and the area under the ROC curve (AUC) value are regarded as the quantitatively evaluation metrics. The ROC curve displays the relationship between the detection probability (DP) and the false alarm rate (FAR), which are given as follows:

$$DP = \frac{N_{cd}}{N_t}, FAR = \frac{N_{fd}}{N}, \tag{14}$$

where $N$ is the total number of pixels in the image, $N_t$ is the total number of anomalous pixels, $N_{cd}$ is the number of correctly detected pixels, and $N_{fd}$ is the number of falsely detected anomalous pixels.

Usually, at the same FAR, the higher DP is, the better the performance the detector can achieve. When the detector performance has a high DP value, the corresponding ROC curve will be located near the upper leftmost corner of the coordinate plane, resulting in a large area under the curve. In addition, the AUC value ranges from 0 and 1, which refers to the area under the ROC curve and is encircled by the ROC curve and the false alarm rate axis. Then, the normalized background-anomaly separation map describes the normalized anomaly score distributions of the background and anomalous pixels which are depicted by a box plot. An outstanding method usually has a high AUC value and a distinct gap between the background box and the anomaly box.

### 4.2. Compared Methods

To evaluate the performance of the proposed RRXEMAP method, we compare it with the following five state-of-the-art methods.

1. GRX: Global RX (GRX) is the representative hyperspectral imagery anomaly detector, which models the background using the whole image.
2. LRX: Local RX (LRX) is the version of RX, which models the background using the local dual-window image. The detection performance of the LRX is sensitive to the sliding double window sizes. In this paper, the inner window size $w_{in}$ ranges from 5 to 11 and the outer window size $w_{out}$ ranges from 7 to 13 to be selected optimally.
3. CRD: The collaborative representation-based anomaly detection (CRD) algorithm for the HSIs assumes that each background pixel can be approximately represented by its surrounding pixels in a local dual window. CRD is also sensitive to the sliding double window sizes; the inner window size $w_{in}$ (ranging from 5 to 11) and the outer window size (ranging from 7 to 13) are selected optimally.
4. LRaSMD: The low-rank and sparse matrix decomposition (LRaSMD) method scores each pixel according to the Euclidean distance between the corresponding sparse component vector and the mean vector of the sparse matrix.
5. LSMAD: The low-rank and sparse matrix decomposition-based Mahalanobis distance (LSMAD) is a typical low-rank and sparse matrix decomposition-based detector, which explores the low-rank prior knowledge of the background pixels and scores each pixel according to the Mahalanobis distance. The parameter of LSMAD refers to the original setting as [62].

### 4.3. Detection Result

Then, we conduct experiments on the above seven datasets to verify the performance of the proposed RRXEMAP method. For the RRXEMAP method, the number of principal components $c = 5$ and the percentage of each dataset is adjusted to optimum.

For the AVIRIS-I dataset, the color detection maps of different methods are presented in Figure 6. It is obvious that the locations of the three airplanes are detected accurately by the proposed RRXEMAP method, and the shapes of the airplanes are also described clearly. Although LRX, CRD and LSMAD can detect the locations of the three airplanes, they also mistakenly identify some background pixels. The GRX and LRaSMD methods fail

to detect the anomalies in this scene. Then, the ROC curves and the corresponding AUC values are shown in Figure 7. From it, we have the following observations. Firstly, it can be seen that the ROC curve of the proposed RRXEMAP algorithm is closer to the top-left corner than the others. Then, the AUC value of RRXEMAP is 0.9940, which is much better than the others.

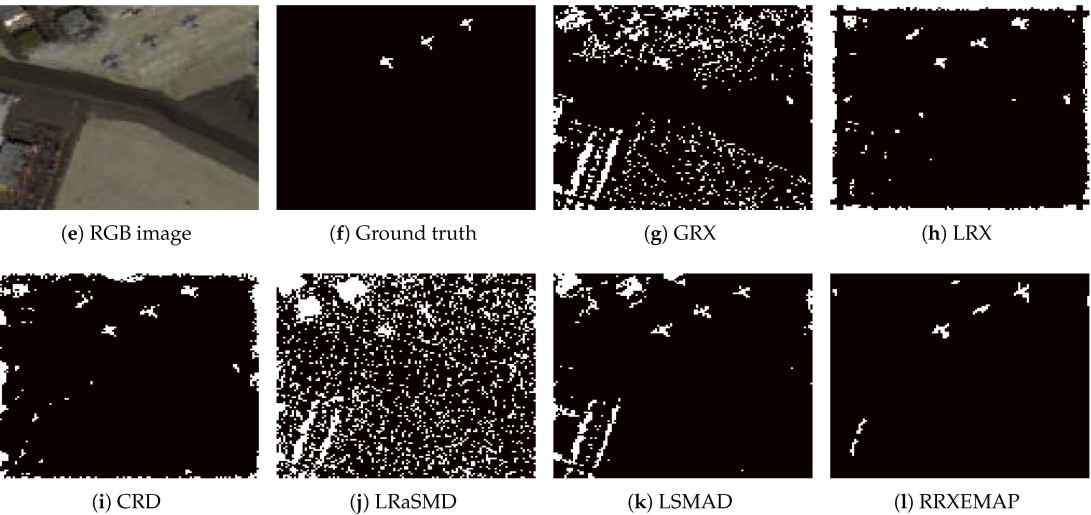

(**e**) RGB image     (**f**) Ground truth     (**g**) GRX     (**h**) LRX

(**i**) CRD     (**j**) LRaSMD     (**k**) LSMAD     (**l**) RRXEMAP

**Figure 6.** Color detection maps obtained by different algorithms for the AVIRIS-I dataset.

For the AVIRIS-II dataset, the color detection maps of different methods are presented in Figure 8. The locations of the three airplanes can be detected accurately by the proposed RRXEMAP method. The locations and shapes of two airplanes can be also detected fairly clearly. The LRX, CRD and LRaSMD detectors fail to detect the anomalies in this scene. The GRX and LSMAD detectors can detect the locations of the three airplanes, while they mistakenly identify some background pixels as anomalies. Figure 9 shows the ROC curves of these detectors and their corresponding AUC values. It can be seen that the ROC curve of the RRXEMAP method is much closer to the top-left corner than the others, and the corresponding AUC value is 0.9790, which is larger than the others.

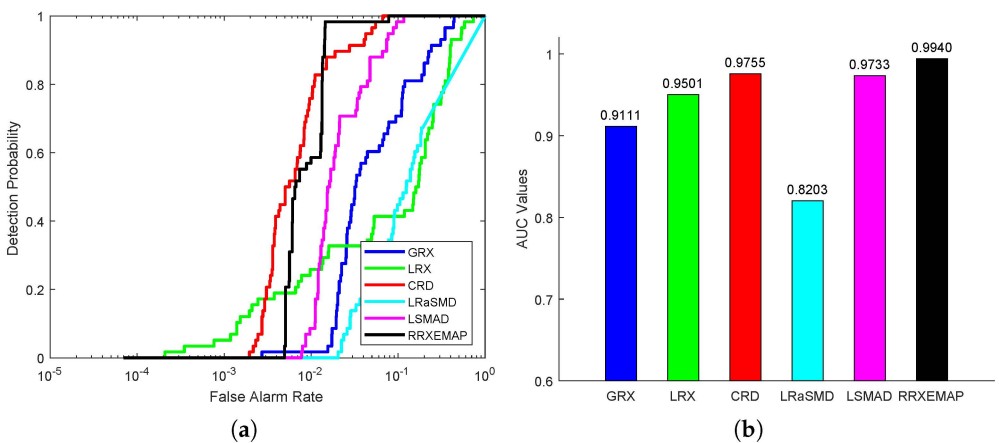

(**a**)

(**b**)

**Figure 7.** Detection accuracy evaluation for the AVIRIS-I dataset. (**a**) ROC curves. (**b**) AUC values.

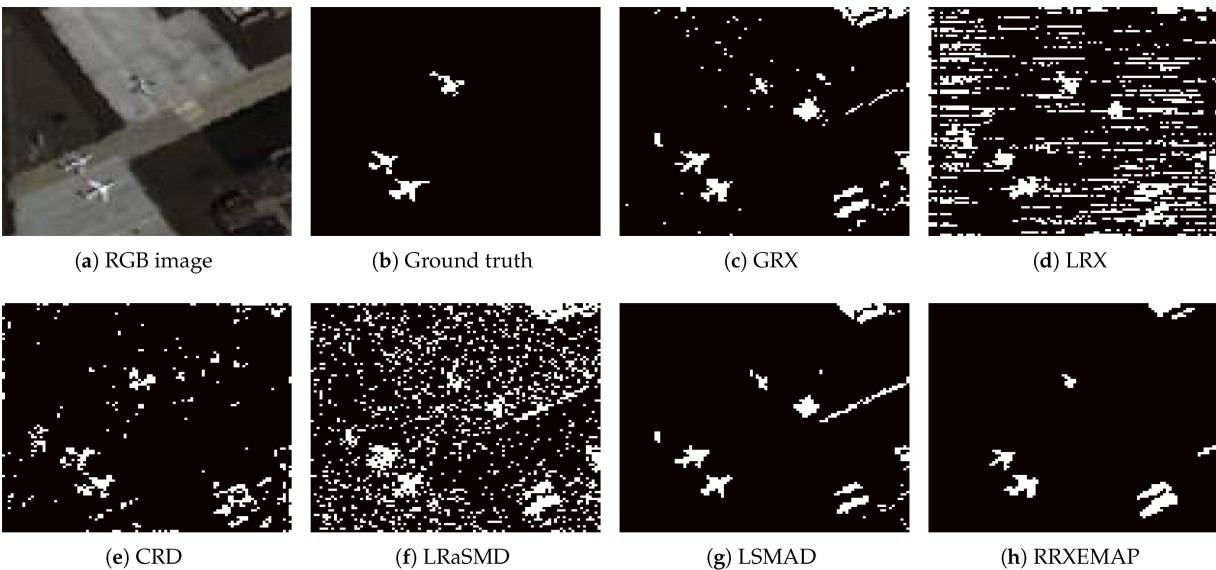

**Figure 8.** Color detection maps obtained by different algorithms for the AVIRIS-II dataset.

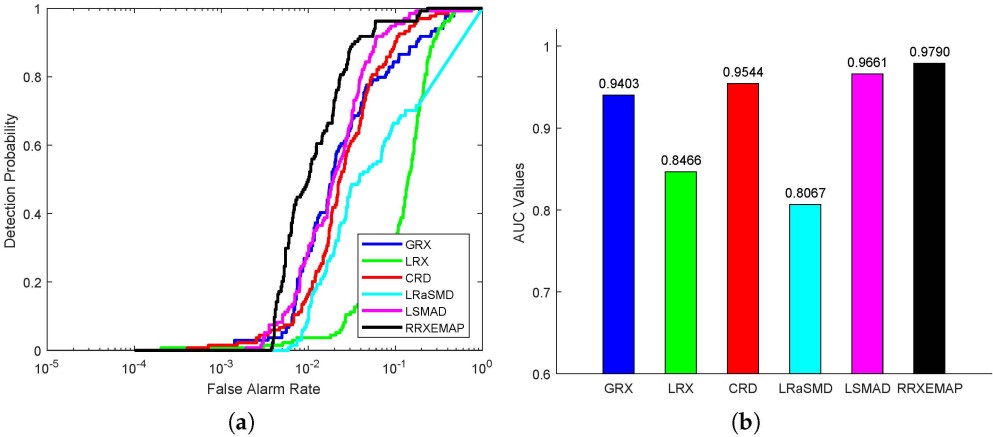

**Figure 9.** Detection accuracy evaluation for the AVIRIS-II dataset. (**a**) ROC curves. (**b**) AUC values.

For the Cri dataset, the color detection maps of different methods are presented in Figure 10. All the detectors (GRX, LRX, CRD, LRaSMD, LSMAD and RRXEMAP) can detect the locations and shapes of the rocks, while the others (GRX, LRX, CRD, LRaSMD and LSMAD) mistakenly identify many background pixels as anomalies. Only the RRXEMAP detector can obtain a cleaner background. Figure 11a shows that the ROC curves of the RRXEMAP method are much closer to the top-left corner than the others in most situations. The AUC value of the proposed RRXEMAP method reaches 0.9943 in Figure 11b, which is 0.0809 higher than the GRX algorithm.

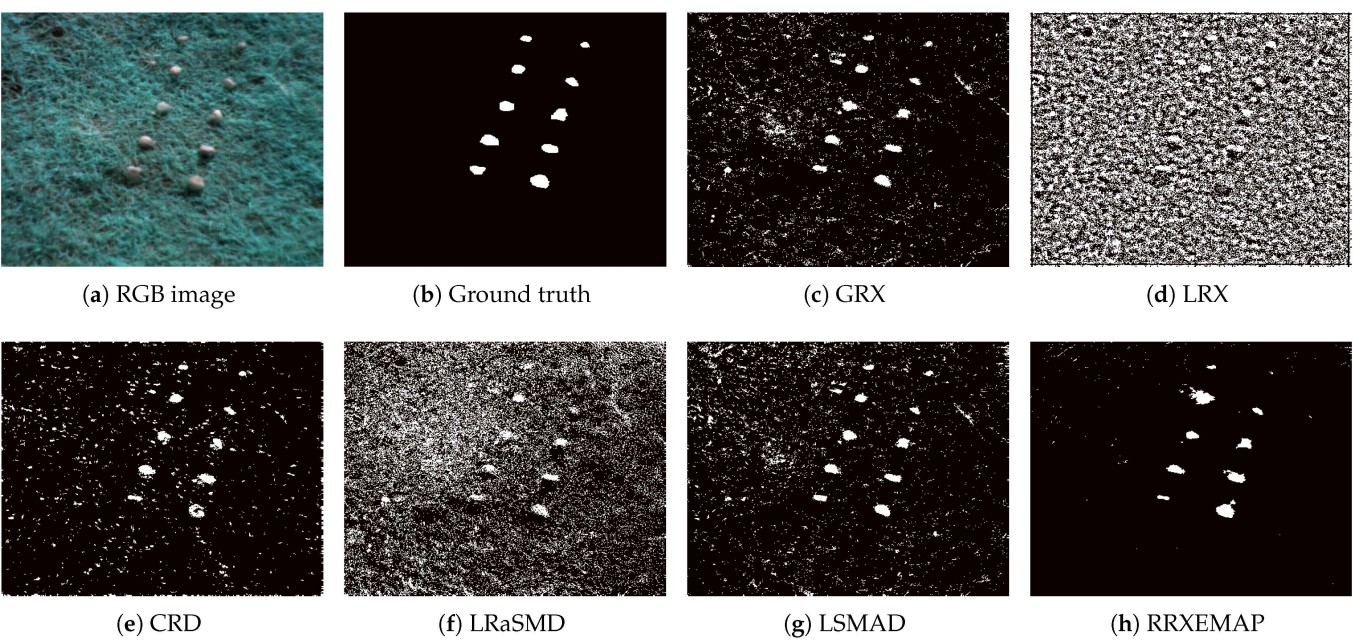

(**a**) RGB image  (**b**) Ground truth  (**c**) GRX  (**d**) LRX

(**e**) CRD  (**f**) LRaSMD  (**g**) LSMAD  (**h**) RRXEMAP

**Figure 10.** Color detection maps obtained by different algorithms for the Cri dataset.

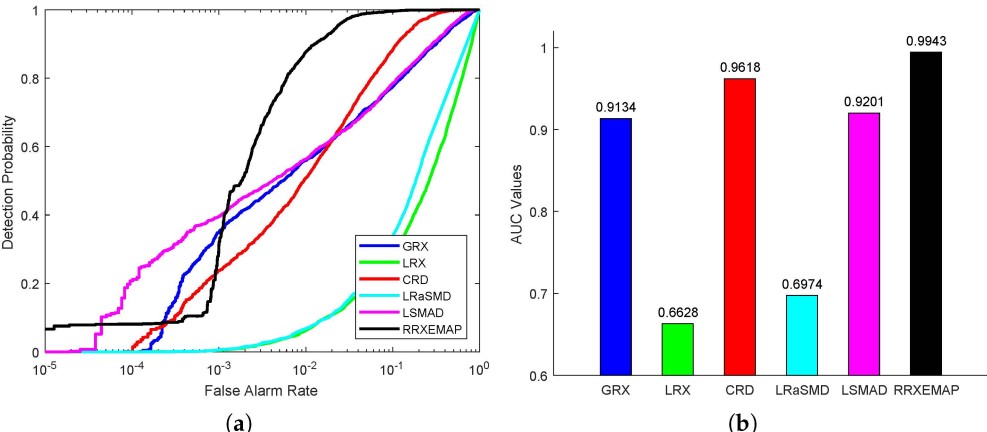

(**a**)

(**b**)

**Figure 11.** Detection accuracy evaluation for the Cri dataset. (**a**) ROC curves. (**b**) AUC values.

For the abu-airport-2 dataset, the color detection maps of different methods are presented in Figure 12. All the detectors (GRX, LRX, CRD, LRaSMD, LSMAD and RRXEMAP) can detect the locations and shapes of the two airports, while the others (GRX, LRX, CRD, LRaSMD and LSMAD) mistakenly identify many background pixels as anomalies. The RRXEMAP detector can obtain a better background. Figure 13a presents the ROC curves of all the methods, from which we can find that the ROC curves of the RRXEMAP method are much closer to the top-left corner than the others in most situations. Figure 13b shows the corresponding AUC value, and the RRXEMAP method of that is 0.9858, which is 0.0.1454 higher than GRX.

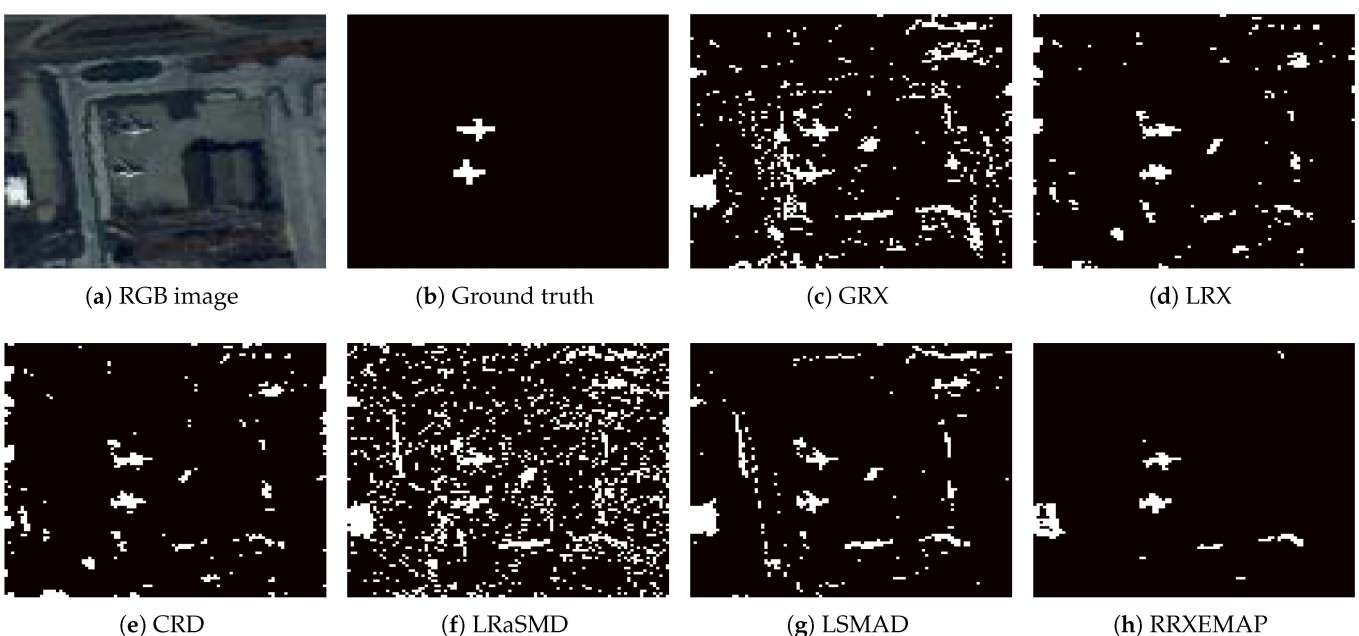

**Figure 12.** Color detection maps obtained by different algorithms for the abu-airport-2 dataset.

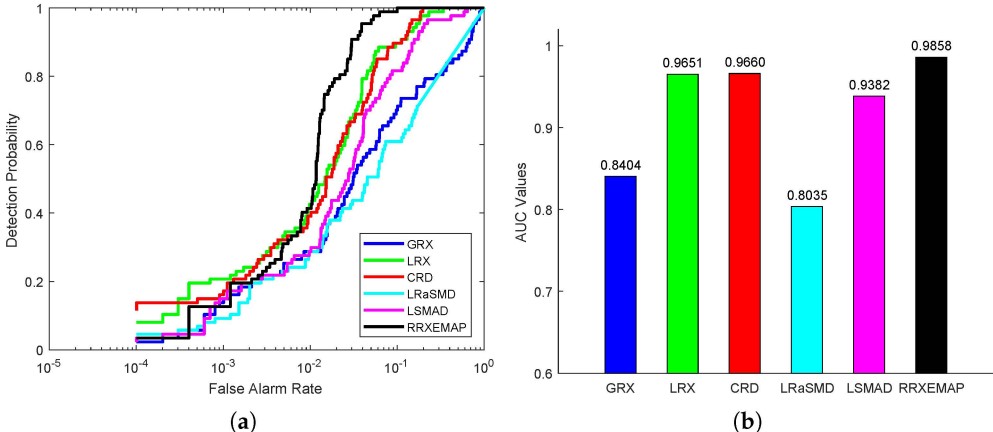

**Figure 13.** Detection accuracy evaluation for the abu-airport-2 dataset. (**a**) ROC curves. (**b**) AUC values.

For the abu-urban-1 dataset, the color detection maps of different methods are shown in Figure 14. All the detectors (GRX, LRX, CRD, LRaSMD, LSMAD and RRXEMAP) can detect the locations and shapes of the anomalies, while the others (GRX, LRX, CRD, LRaSMD and LSMAD) mistakenly identify some background pixels as anomalies. Figure 15a shows the ROC curves of all detectors, from which we can find that the ROC of the RRXEMAP method is much closer to the top-left corner than the others in most situations. In addition, the corresponding AUC value of the RRXEMAP method reaches 0.9961 in Figure 15b, which is higher than the others.

For the abu-urban-3 dataset, the color detection maps of different methods are presented in Figure 16. The RRXEMAP algorithm can obtain a clear result, while the others (GRX, LRX, CRD, LRaSMD and LSMAD) mistakenly identify many background pixels as anomalies. Figure 17a shows that the ROC curves of the RRXEMAP method are much closer to the top-left corner than the others in most situations. The corresponding AUC value of the proposed RRXEMAP method reaches 0.9873 in Figure 17b, which is 0.036 higher than the GRX approach.

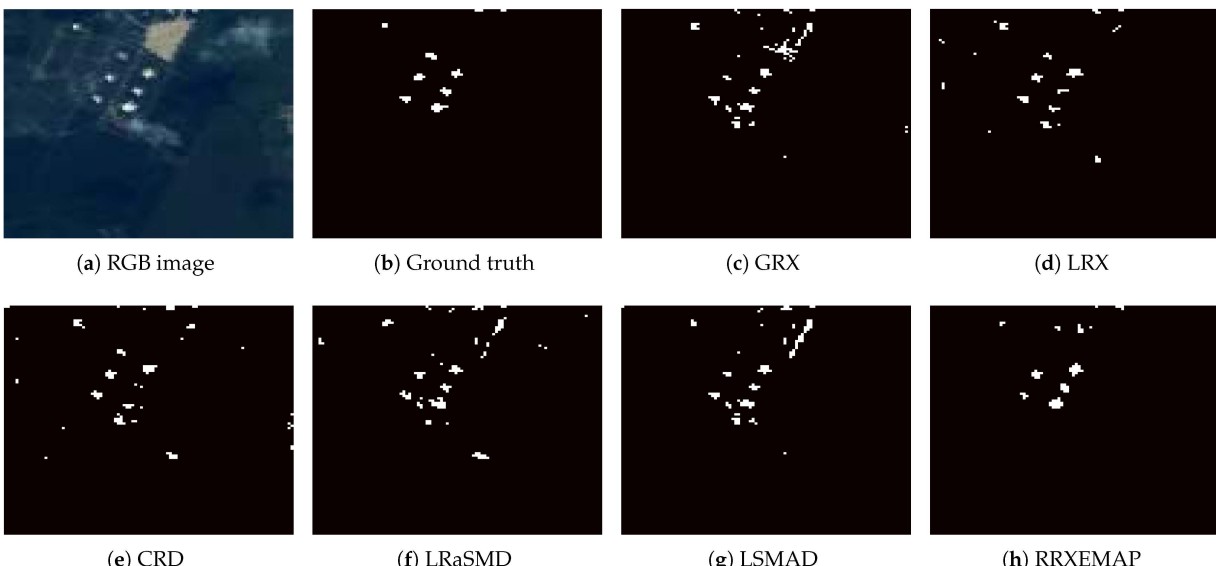

**Figure 14.** Color detection maps obtained by different algorithms for the abu-urban-1 dataset.

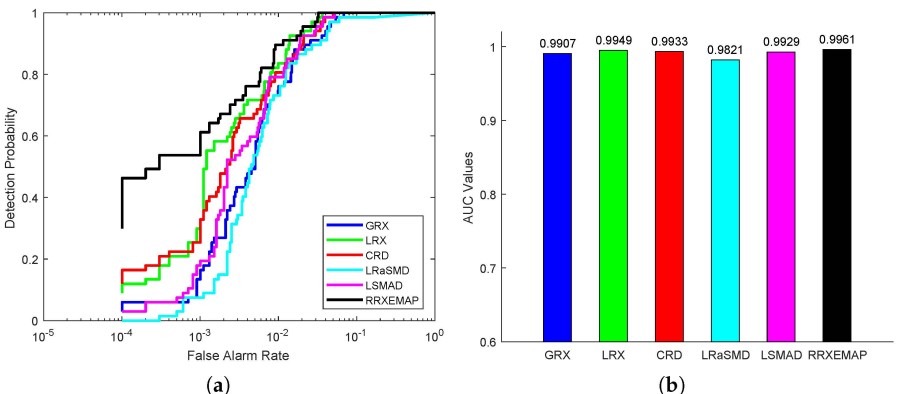

**Figure 15.** Detection accuracy evaluation for the abu-urban-1 dataset. (**a**) ROC curves. (**b**) AUC values.

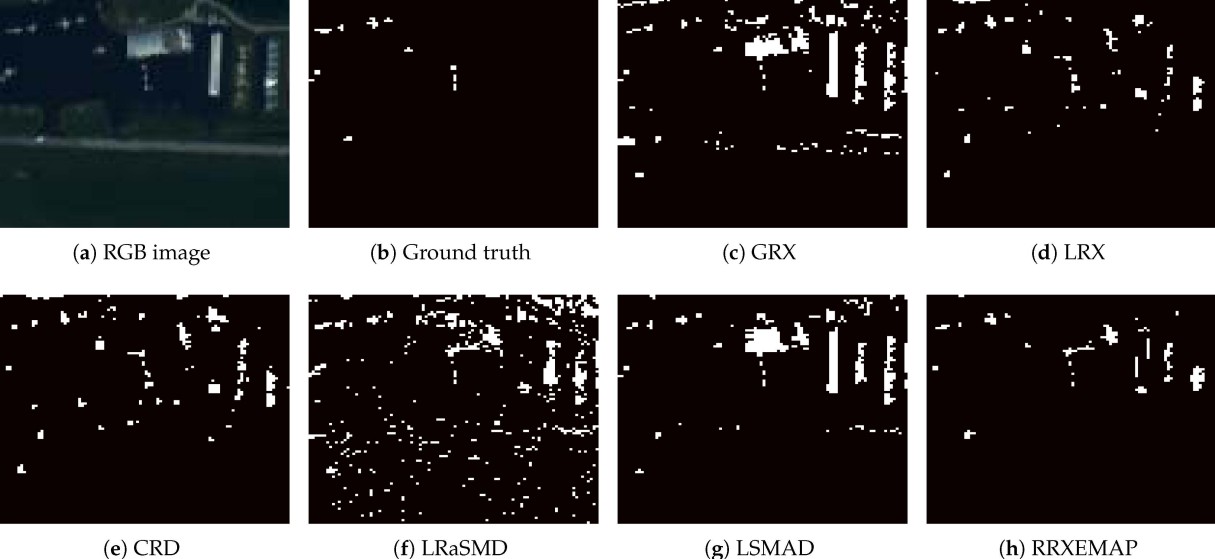

**Figure 16.** Color detection maps obtained by different algorithms for the abu-urban-3 dataset.

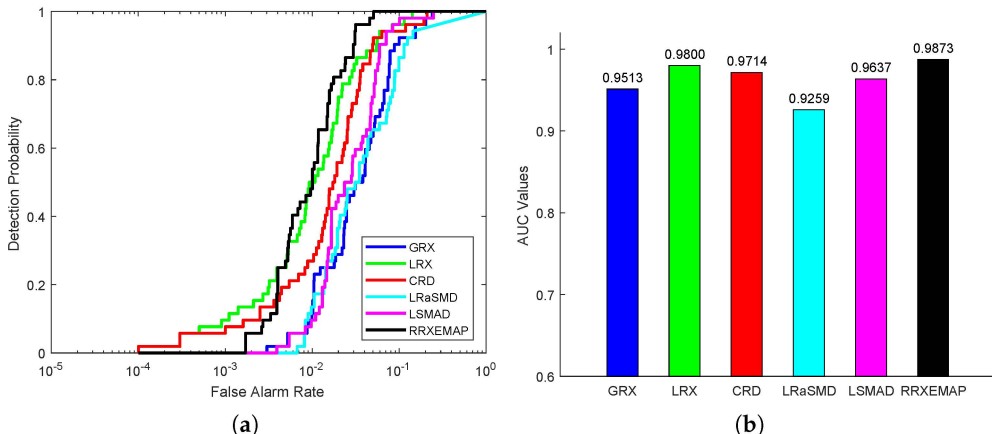

**Figure 17.** Detection accuracy evaluation for the abu-urban-3 dataset. (**a**) ROC curves. (**b**) AUC values.

For the Salinas-simulate dataset, the color detection maps of different methods are presented in Figure 18. GRX, LSMAD and RRXEMAP can detect the locations and shapes of the anomalies, while LRX, CRD and LRaSMD cannot. Compared with the GRX and LSMAD algorithms, RRXEMAP has a better ability to detect anomaly with higher accuracy. The GRX and LSMAD models mistakenly identify more background pixels as anomalies than the RRXEMAP algorithm. Figure 19a shows that the ROC curves of the RRXEMAP method are much closer to the top-left corner than the others in most situations. The AUC value of the proposed RRXEMAP method reaches 0.9992 in Figure 19b. These results illustrate the superiority of the RRXEMAP detector.

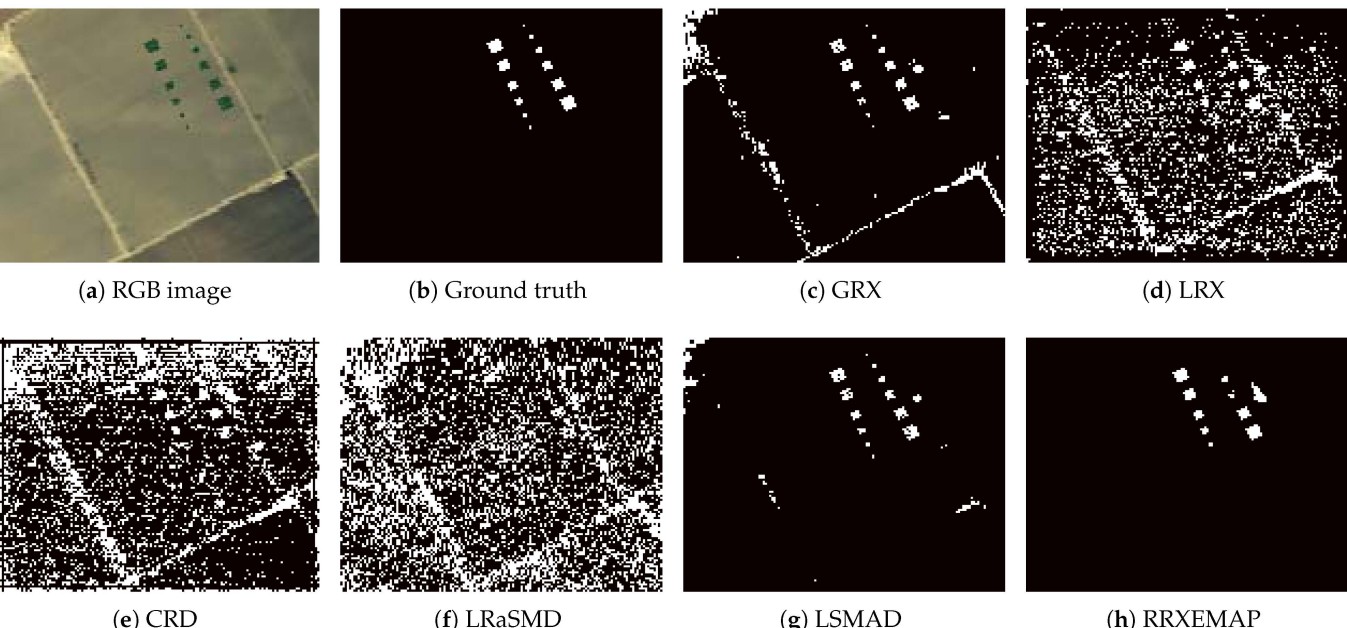

**Figure 18.** Color detection maps obtained by different algorithms for the Salinas-simulate dataset.

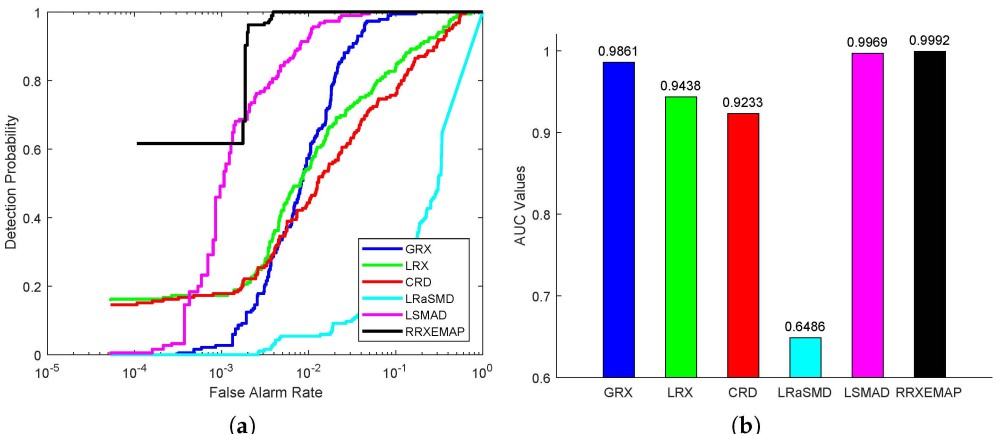

**Figure 19.** Detection accuracy evaluation for the Salinas-simulate dataset. (**a**) ROC curves. (**b**) AUC values.

*4.4. Background Purity*

In this part, we conduct experiments to analyze the importance of parameter *p*. Taking AVIRIS-I and Cri datasets as examples, the parameter *p* varies from 40% to 100%, whose corresponding results are shown in Figure 20. It can be found that the AUC value of RRXEMAP changes as *p* changes. On the AVIRIS-I dataset, the AUC of the proposed RRXEMAP ranges from 0.9566 to 0.9940 for different percentages, which is a difference of 0.0374. Therefore, the parameter *p* should be adjusted carefully to make the RRXEMAP method achieve better performance.

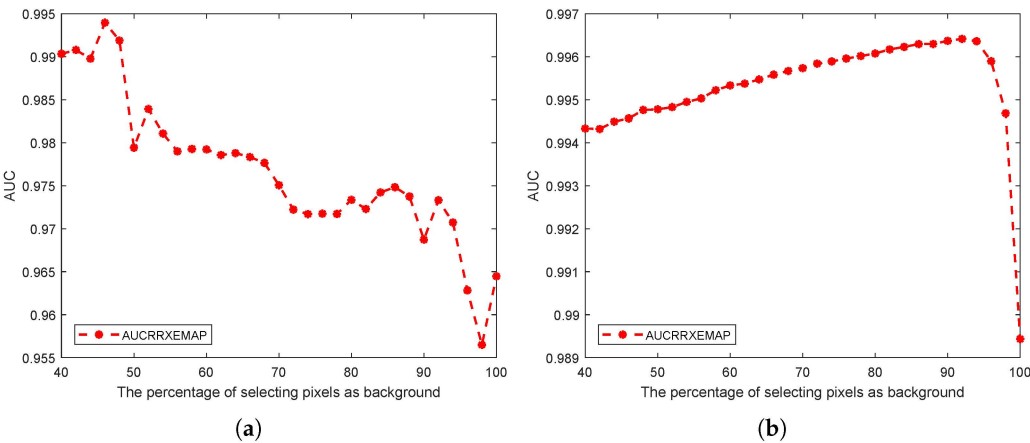

**Figure 20.** Effect of the percentage of pixels selected as background on each dataset. (**a**) AVIRIS-I. (**b**) Cri.

**5. Conclusions**

In this paper, we proposed a novel HAD algorithm called Recursive RX with Extended Multi-Attribute Profiles (RRXEMAP). Firstly, the extended multi-attribute profiles (EMAP) method is utilized to extract the spatial features of the hyperspectral image, which can make use of the spatial information of HSI to improve the detection performance. Then, a simple background purification method is adopted by using the RX detector to remove the pixels that are most likely to be anomalies. Finally, the RX method is used again to complete the final anomaly detection. Experimental results on six real hyperspectral datasets and a synthetic dataset demonstrate the superiority of the proposed RRXEMAP method. However, the proposed method is time consuming, as it is a linear combination of the EMAP and RX algorithms. The complexity of the proposed method should be further reduced in the future for the other applications, such as real-time or near-real-time scenarios.

**Author Contributions:** Conceptualization, F.H. and H.H.; methodology, H.H.; software, S.Y.; investigation, Y.D.; resources, Z.S.; data curation, J.Z.; writing—original draft preparation, F.H.; writing—review and editing, Y.Z.; funding acquisition, Z.S. All authors have read and agreed to the published version of the manuscript.

**Funding:** This research is supported by Shaanxi Natural Science Foundation Grant 2023-JC-QN-0052 and 2023-JC-QN-0027, the project 2021-JCJQ-JJ-0424 and SYS-ZZ-202207006.

**Data Availability Statement:** Publicly available data sets were analyzed in this study, which can be found here: http://www.ehu.eus/ccwintco/index.php?title=Hyperspectral_Remote_Sensing_Scenes, accessed on 30 November 2022.

**Acknowledgments:** We thank the editors and reviewers for their impressive work.

**Conflicts of Interest:** The authors declare no conflict of interest.

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
