# Peer review of "Recursive RX with Extended Multi-Attribute Profiles for Hyperspectral Anomaly Detection"

_remotesensing, doi:10.3390/rs15030589_

Round 1

Reviewer 1 Report

In this manuscript, Recursive RX with Extended Multi-Attribute Profiles (RRXEMAP) for Hyperspectral Anomaly Detection (HAD) is proposed. The model is easy to implement. Firstly,  the spatial features of the hyperspectral image are extracted using the extended multi-attribute profiles (EMAP) approach. Then, a straightforward background purification technique is used by removing the pixels that are most likely to be anomalous using the RX detector. Finally, the final anomaly detection is finished by reusing the RX approach. The results are persuasive. For example, the AUC value is 0.9790 on AVIRIS-II dataset, 0.0129 higher than the second one. This model is interesting and there are some convincing experimental results. However, there are some suggestions that you can correct carefully.

1. Grammar and typo errors are shown in the manuscript frequently. For example, In line 352, this sentence “illustrated in Fig.ref. Fig.21 using the AVIRIS-I and Cri datasets as examples” , please check it carefully. In line 362, this sentence  “utilise the spatial data of the HSI to improve the detection performance”, should be corrected.

2. More references about anomaly detection model are needed.

3. In the Experiments section, many attentions on the results are paid. Please explain the reason of your results. The result analysis should be more detailed.

Author Response

Thanks for your comments. The response is in the attachment. Please see the attachment.

Reviewer 2 Report

In this manuscript, the authors proposed a Hyperspectral Anomaly Detection (HAD) method, named Recursive RX with Extended Multi-Attribute Profiles (RRXEMAP). It is meaningful because the proposed RRXEMAP is easy to realize and improves detection accuracy. And the proposed RRXEMAP model extracts the HSI spatial information and achieves background purification, which is helpful for HAD. However, to help improve the quality of this manuscript and make it outstanding among the published works, I have provided the following comments. 

1. Authors should pay attention to the language and format of the manuscript. For example, the size of figure.2 is too large. Please adjust it to make the structure more beautiful. In line 161, the sentence “which can extract spatial features to more effectively use the spatial information” is too wordy, please simplify it.

2. During the past years, many scholars have dealt with this topic. What is the novelty?

3. The abstract should be further refined. For example, the parameter selection part in the paper is not reflected in the abstract. Besides, please add the numeric result of the experimental part in the abstract.

4. There are some grammar errors in the paper and proofreading needs to be conducted.

5. The main contribution shall be given.

6. The authors need to improve the language of the paper and add some more recent state-of-the-art works.

Author Response

Thanks for your comments. Please see the attachment of our response.

Reviewer 3 Report

In this work, authors have proposed the Recursive RX with Extended Multi-Attribute Profiles (RRXEMAP) method for taking into account the spectral properties of HSI in HAD problem. The work seems to be interesting, however, I have the following suggestions/queries for the authors for possible improvements.

i) Do review the article for possible grammatical or other mistakes.

For example, 

a) In the abstract, the acronym HSI is used even before its definition.

b) Repetition of text in lines 57-63.

ii) Authors say that "The detailed description of GRX is shown in Algorithm 1." However, Algorithm 1 is vague and should be updated with proper descriptions of different GRX stages. Also, the question is do authors really need to present algorithm 1 ? since it is already available in the literature and authors are just reviewing it here.

iii) How about the complexity introduced by the proposed algorithm? Since the proposed algorithm is the recursive version of a baseline RX method, will the complexity become an issue? Anomaly detection can have some latency-sensitive applications i.e., UAV-based systems. Will the proposed method be useful for such real-time or near-real-time scenarios?

Author Response

(The authors gave the same response as above.)

Round 2

Reviewer 2 Report

I accept the paper for publication in the current version.

Author Response

Thanks for your agreement. We add another author as his contribution of our revised manuscript.

Reviewer 3 Report

Thank you for providing adequate responses to the previous queries. I suggest you add the complexity analysis part in the article (a refined version of your response to my 3rd question from the previous round) to provide the pros/cons of the proposed methods and further motivate the research in that direction. Apart from that, I don't have any further suggestions/questions for this work. 
